# Selective Unlearning via Representation Erasure Using Domain Adversarial Training

**Nazanin Mohammadi Sepahvand**[1][*]    **Eleni Triantafillou**[2]    **Hugo Larochelle**[2]    **Doina Precup**[1,2]
**James J. Clark**[1]    **Daniel M. Roy**[3]    **Gintare Karolina Dziugaite**[1,2]
[1]McGill University, Mila, Canada    [2]Google DeepMind    [3]University of Toronto, Canada

## Abstract

When deploying machine learning models in the real world, we often face the challenge of "unlearning" specific data points or subsets after training. Inspired by Domain-Adversarial Training of Neural Networks (DANN), we propose a novel algorithm, SURE, for targeted unlearning. SURE treats the process as a domain adaptation problem, where the "forget set" (data to be removed) and a validation set from the same distribution form two distinct domains. We train a domain classifier to discriminate between representations from the forget and validation sets. Using a gradient reversal strategy similar to DANN, we perform gradient updates to the representations to "fool" the domain classifier and thus obfuscate representations belonging to the forget set. Simultaneously, gradient descent is applied to the retain set (original training data minus the forget set) to preserve its classification performance. Unlike other unlearning approaches whose training objectives are built based on model outputs, SURE directly manipulates the *representations*. This is key to ensure robustness against a set of more powerful attacks than currently considered in the literature, that aim to detect which examples were unlearned through access to learned embeddings. Our thorough experiments reveal that SURE has a better unlearning quality to utility trade-off compared to other standard unlearning techniques for deep neural networks.

## 1 Introduction

In machine learning, the principle of unlearning is essential for selectively removing the influence of specific data points from a trained model. The gold standard is exact unlearning, which demands that the model behaves as if it never encountered the data to be forgotten (the *"forget set"*). Exact unlearning, however, presents significant challenges in deep learning models due to their non-linear learning dynamics. A naive implementation of exact unlearning may require retraining the model on a subset of the data that excludes the forget set, which is often prohibitively costly. As a result, the focus has shifted towards approximate unlearning methods, which aim to mimic the behavior of an oracle model - a model retrained from scratch without the forget set.

Traditional approximate unlearning methods primarily focus on aligning the model's output after unlearning with that of the oracle model. However, we hypothesize, and show empirically, that it may be possible for an unlearning algorithm to appear to have forgotten, if considering only information in the output space, while information about the forget set may in fact remain hidden in intermediate layers and may be recoverable. This is an important setting to consider in case an attacker gains white-box access to the model weights. Motivated by this observation, we study algorithms and metrics that operate directly in the representation space. Interestingly, by visualizing the embedding space (where the model represents data internally), we in fact observe that successful unlearning is often correlated with representations closely resembling the oracle's, suggesting that directly manipulating representations during unlearning is also a promising approach for designing improved algorithms. Conversely, a mismatch in representations can potentially expose the model to a type of *membership inference attacks* (MIAs), which aim to identify whether a specific data point was part of the training data.

---

[*]Correspondence to: sepahvan@mila.quebec, gkdz@google.com

Motivated by these findings, we introduce SURE (Selective Unlearning via Representation Erasure), a novel unlearning method that shifts the focus from the output space to the representation space. SURE aims to adjust the model's representations to closely match those of the oracle, effectively erasing the influence of the forget set.

Through extensive empirical evaluation, we demonstrate that SURE not only achieves a superior trade-off between unlearning effectiveness and model utility but also exhibits reduced vulnerability to representation-space MIAs.

**Contributions.**

- **Enhanced Unlearning Quality**: SURE demonstrates superior unlearning quality as assessed by standard Membership Inference Attacks (MIAs) and the accuracy difference between the unlearned model and the oracle model. This is further corroborated by comparing the accuracy of unlearned and oracle models on retain, forget, and test sets across various unlearning scenarios, including random, partial class, and class unlearning, as well as scenarios with a forget set containing outliers from different distributions.

- **Improved Stability in Unlearning**: Addressing a significant limitation of existing unlearning methods, SURE offers enhanced stability. Unlike methods where minor changes in hyperparameters or forget set size can lead to substantial variations in results, SURE demonstrates consistent performance across different forget set sizes, and when averaging results over multiple epochs, ensuring reliability in real-world scenarios.

- **Reduced Vulnerability to Representation Space Attacks**: SURE effectively mitigates vulnerabilities to MIAs targeting the representation space, a weakness exhibited by some existing methods. This is achieved through the proposed representation erasure technique and is visualized using t-SNE plots, showcasing the distinct advantage of SURE in enhancing privacy.

## 2 PRELIMINARIES

### 2.1 DANN: DOMAIN-ADVERSARIAL NEURAL NETWORK

We start by describing the key framework that we adopt, DANN, as introduced by Ganin et al. (2016). The authors proposed a simple yet powerful idea for unsupervised domain adaptation with the goal of improved performance under domain shifts. The main insight incorporated into DANN is to force the neural network to learn features (representations) that are: (1) discriminative for the task; and (2) domain-invariant, meaning that the features should not reveal whether the input data came from the source domain or the target domain.

Figure 1 provides a schematic overview of the DANN framework. DANN comprises three key components, each with their own parameterizations:

- *Feature extractor module* $G_f(\cdot; \theta_f)$: In this work, the backbone of a ResNet-18 is used for this module.

- *Label classification module* $G_y(\cdot; \theta_y)$: This module takes feature extractor outputs, and maps them to labels. Here, this is a simple fully-connected classification layer with $K$ neurons, where $K$ is the number of classes.

- *Domain regressor module* $G_d(\cdot; \theta_d)$: This is a binary classification component modelling the probability that a given input is from the source vs target domain. In this work we use a two-layer fully connected neural networks for this module.

Take a feature extractor module trained on some source data, parameterized by $\theta_f$. Informally, the training process for DANN simultaneously optimizes the following: domain regressor module parameters $\theta_d$ are trained to distinguish between the source and target features (discrimination task); the feature extractor $\theta_f$ is fine-tuned to fool the domain regressor; the label classification module $\theta_y$ is trained to maximize performance on the source data (classification task). To fine-tune the feature extractor, a *Gradient Reversal Layer* (GRL) is inserted between the feature extractor and the domain regressor module. GRL reverses the sign of the gradients flowing from the domain regressor

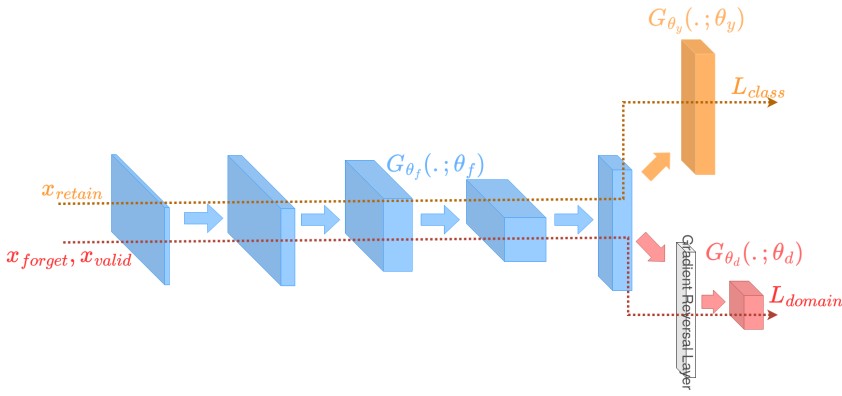

Figure 1: A schematic representation of SURE. The feature extractor is illustrated in blue, classification module in yellow, and domain regressor in red, with the gradient reversal layer in between the feature extractor and domain regressor. Note that a similar schematic representation also captures DANN but with different input distributions.

back to the feature extractor while, during forward propagation, the GRL functions as an identity transformation. This forces the feature extractor to learn features that make it difficult for the domain regressor to distinguish between the source and target domains.

## 2.2 UNLEARNING SETTING

Let $X$ denote the input space, and $Y = \{0, \ldots, K-1\}$ the set of $K$ possible labels. The source training dataset, $D = (x_i, y_i)_{i=1}^{N}$, comprises $N$ i.i.d. samples drawn from the data distribution. During unlearning, $D$ is partitioned into two mutually exclusive subsets: a *forget set* $D_{\mathrm{f}}$, containing samples for which the model must "unlearn" information; and *retain set* $D_{\mathrm{r}} = D \setminus D_{\mathrm{f}}$.

In addition, we assume access to a validation set $D_{\mathrm{val}}$. For simplicity of presentation, this validation set is assumed to be of the same size as the forget set. Samples in the validation set were not seen during the original training, may be unlabeled, and are drawn from a target distribution of interest (which could be the same as the source distribution). The set to which each sample $x$ belongs is discerned through the *domain labels* $d \in \{0, 1\}$ (the class label here is irrelevant or may not be given). This label is set to one for the forget set and zero for the validation set.

We use $D_{\mathrm{val}}^{x}$ and $D_{\mathrm{f}}^{x}$ to denote only the input space samples without the labels from the validation and forget sets respectively; $D_{\mathrm{test}}$ refers to the test set.

## 3 UNLEARNING DEFINITIONS AND ASSOCIATED METRICS

### 3.1 FROM CLASSICAL NOTIONS TO REPRESENTATION-FOCUSED UNLEARNING DEFINITION

A perfect unlearning algorithm would ensure a model behaves exactly as if the removed data never existed. This notion can be formalized by drawing parallels to differential privacy (Dwork, 2006). We begin by defining a measure of closeness between distributions: two distributions $\mu, \nu$ are said to be $(\epsilon, \delta)$-*close* if $\mu(B) \leq e^{\epsilon}\nu(B) + \delta$ and $\nu(B) \leq e^{\epsilon}\mu(B) + \delta$ for all measurable events $B$. With this notion of closeness, we can formally define an unlearning algorithm:

**Definition 3.1.** *An unlearning algorithm $\mathcal{U}$ is an $(\epsilon, \delta)$-unlearner for a learning algorithm $\mathcal{A}$, and training set $D$ if, for all subsets $D_{\mathrm{f}} \subseteq D$ of fixed size, the distribution of $\mathcal{A}(D \setminus D_{\mathrm{f}})$ and $\mathcal{U}(\mathcal{A}(D), D_{\mathrm{f}})$ are $(\epsilon, \delta)$-close.*

This definition captures the essence of unlearning by quantifying the difference between a model trained without the forget set and a model that underwent unlearning the forget set. However, this raises a crucial question: on what output distribution are we measuring this difference?

In Definition 3.1, the (potentially randomized) learning algorithm $\mathcal{A}$ takes a training set $D$ and produces a distribution over outputs. A common approach is to consider $\mathcal{A}(\cdot)$ as returning a distribution over loss values. If $\theta$ represents the learned weights (which are random variables due to the stochastic nature of training), and $f(\cdot; \theta)$ is the neural network parameterized by $\theta$, then the distribution

considered in previous work is the average loss over the forget set: $L(D_f^y, f(D_f^x; \theta))$. Alternatively, one could consider the individual losses of each input in $D_f$. This approach to evaluating unlearning quality was used in the NeurIPS'23 competition (Triantafillou et al., 2024) and in defining the LiRA membership inference attack (Carlini et al., 2022) which was adapted to the context of unlearning in Hayes et al. (2024).

Another possibility is to directly compare the distributions of the learned weights ($\theta$'s). However, the high dimensionality and permutation symmetries of neural networks make this comparison challenging, and it remains largely unexplored. A third option involves comparing lower-dimensional representations learned by the network, such as penultimate layer activations, which are usually used for the so-called conjugate kernel computation (e.g., (Hu & Huang, 2021)).

Due to the data processing inequality, we know that the information signal, and thus the closeness between $\mathcal{A}(D \setminus D_f)$ and $\mathcal{U}(\mathcal{A}(D), D_f)$ (as measured by $(\epsilon, \delta)$), cannot be increased when moving from comparing $\theta$'s to comparing $L(D_f^y, f(D_f^x; \theta))$. Therefore, comparing unlearned models to retrained models based on losses might provide a more optimistic view of unlearning performance relative to comparisons based on weights.

In our work, we go beyond comparing losses and evaluate the closeness between $\mathcal{A}(D \setminus D_f)$ and $\mathcal{U}(\mathcal{A}(D), D_f)$ using various outputs, including learned weights and internal representations, as described in Section 3.2.

## 3.2 UNLEARNING METRICS

**Efficiency**: The primary goal of approximate unlearning is to efficiently and effectively remove the influence of forget samples while achieving performance comparable to exact unlearning. To measure computation efficiency, we report the *Relative Run Time (RRT)*, calculated as the ratio of the time taken to unlearn the samples to the time required for an identical model to retrain from scratch on $D_r$ (oracle). To assess effectiveness, we compare the unlearned model to the oracle using various metrics to determine how closely it mimics the oracle's behavior. Specifically, we use the following metrics:

**Accuracy Gap**: We compare the unlearned model with the oracle by evaluating the an accuracy gap, i.e., difference in accuracy between the two models, on different subsets of data, each measuring distinct aspects of unlearning (Liu et al., 2024). *Remaining Accuracy (RA)*, defined as retain set accuracy gap, assesses the fidelity of the unlearning method; *Utility*, defined as test set accuracy gap, reflects generalization ability; and *Unlearning Accuracy (UA)*, defined as forget set accuracy gap, measures the effectiveness of unlearning. Additionally, for class unlearning scenarios, *Class Test Accuracy (CTA)* measures test set accuracy gap for the specific class undergoing partial or complete unlearning.

**Canary Unlearning**: To further evaluate the effectiveness of our unlearning method, we consider the concept of unintended memorization. Carlini et al. (2019) observed that neural networks may inadvertently memorize out-of-distribution (OOD) training samples, even though these samples, by definition, are unrelated to the target distribution and do not contribute to improving model accuracy. This phenomenon can inadvertently reveal the presence of specific samples in the training data. Specifically, we replace 50 samples from the forget set with OOD samples, referred to as "canaries", as described in (Carlini et al., 2019), creating a mixed forget set comprising both in-distribution (IN) and OOD samples. The original model (before unlearning) is trained using this modified training set, which serves as the starting point for all unlearning methods. This experimental setup allows us to investigate the robustness of our unlearning method in handling unintended memorization and effectively removing the influence of these OOD samples.

**Membership Inference Attack**: Membership Inference Attacks (MIAs) serve as a crucial tool for evaluating the effectiveness of unlearning techniques by assessing the extent to which a model has truly "forgotten" the data it was instructed to unlearn. In the unlearning literature, various versions of MIAs have been introduced (Carlini et al., 2022; Golatkar et al., 2021; Goel et al., 2022b). In this work we use two of the commonly employed ones. The first version (MIA-I) involves training a binary classifier to distinguish between samples from the *forget set*, $D_f$, and those from the test set, $D_{\text{test}}$, (Kurmanji et al., 2024). This classifier relies on the loss values produced by the unlearned model when presented with these samples. A higher accuracy of this classifier indicates that the

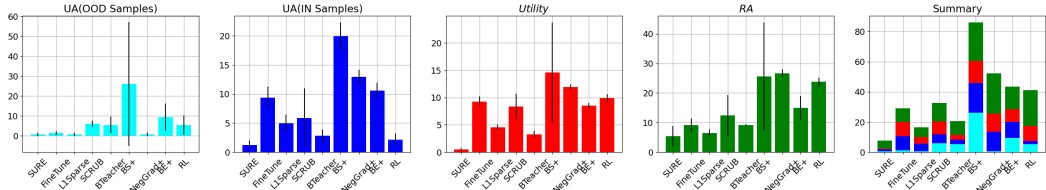

Figure 2: Canary outlier experiment results. The forget set, $D_f$ consists of 1000 samples, 50 of which are completely scrambled (OOD samples), while the remaining samples are randomly selected from the training set (IN samples).Shorter bars reflect narrower accuracy gaps, suggesting that the performance is closer to that of the oracle. The summary plot (far right) displays all the plots from the first four panels stacked on top of each other.

unlearned model still retains information about the forget set, making it susceptible to membership inference. The second version of MIA (MIA-II) trains a classifier to differentiate between *training*, $D$, and test samples, $D_{\text{test}}$, based on the model's confidence in its predictions (Liu et al., 2024; Jia et al., 2023; Song & Mittal, 2021). This classifier is then used to classify forget samples, $D_f$, during inference. A higher percentage of forget samples classified as test samples suggests that the model has successfully unlearned the forget set, as its confidence values resemble those of samples it has never seen before. For both types of MIAs, the performance of the unlearned model is compared to that of the oracle model (a model retrained from scratch without the forget set) to assess the effectiveness of the unlearning technique.

Membership Inference Attacks (MIA) are traditionally conducted as black-box attacks, where the attacker relies solely on the model's outputs to determine whether a specific data point was included in the training set. In this work, we extend MIA to white-box attacks (Sablayrolles et al., 2019), assuming the attacker has access to the model's internal representation space, potentially through its weights. This is achieved by implementing modified versions of the two discussed MIAs. Specifically, the modified version of MIA-I involves training a k-Nearest Neighbors (KNN) classifier for a binary classification task, aimed at distinguishing between the forgotten set, $D_f$, and the test set, $D_{\text{test}}$, based on their embedding representations. Conversely, the modified version of MIA-II trains a KNN classifier to distinguish between train samples, $D$, and test samples, $D_{\text{test}}$, based on their representations, subsequently applying this trained KNN to classify the forget samples, $D_f$. While traditional black-box attacks often use logistic regression, we opt for KNN in the embedding space. This choice is motivated by the higher dimensionality of the embedding space compared to the output space, as well as the potentially small size of the forget set, making a non-parametric classifier more suitable. For brevity, we refer to MIA in Representation Space as **MIARS**.

**Weight Space Comparisons**: To assess the post-unlearning representation in the embedding space, one effective method is to compare the weights and/or activations of the penultimate layer to those of the oracle. However, directly comparing these values is complicated by the *permutation invariance* property (Ainsworth et al., 2022) of neural networks, which arises from the permutation symmetries of the neurons in each layer (Brea et al., 2019). As noted by Nielsen, multiple arrangements of weights can represent the same underlying function, making it challenging to draw meaningful comparisons between the weights of two separately trained networks (Hecht-Nielsen, 1990). To address this issue, we first align the weights of the unlearned model and the oracle using Git Re-Basin, a matching algorithm (Ainsworth et al., 2022) designed to find a set of permutations that minimizes the distance between the two sets of weights/activations. Weights are then compared using Euclidean distance.

## 4   SURE: SELECTIVE UNLEARNING VIA REPRESENTATION ERASURE

Most networks designed for classification tasks can be viewed as consisting of two primary modules: the feature-extractor ($G_f(\cdot; \theta_f)$) and the classifier ($G_y(\cdot; \theta_y)$). Our proposed unlearning-enabled network, SURE, can be viewed as an extension of the original architecture, augmented with an additional domain regressor module during the unlearning process. *Unlike in DANN, however, in SURE the domain regressor will be aiming to distinguish between the forget set, and a validation set that the original model never trained on (instead of the source and target domains).* In our

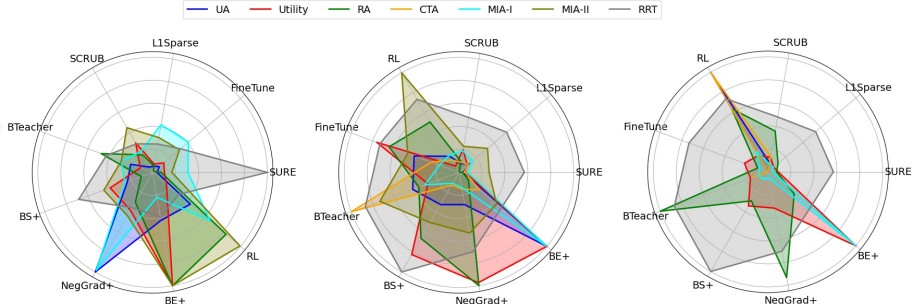

Figure 3: Results for random, partial class, and full class unlearning. For each benchmark, the plot displays the absolute value of the gap between the unlearned model and the oracle, so the ideal is to be as close to the center as possible. The forget set size is 1000 samples for random unlearning and 100 for partial unlearning. Results are averaged over 5 runs, each with a distinct forget set. To enhance the visual clarity of the plot, we normalized each metric's values to the maximum value.

experimental framework, the validation set comes from the same distribution as the forget set. The primary goal of the domain regressor module is to assist the network in unlearning in the feature space. This is accomplished by encouraging the network to change learned features in a way that would be discriminative for the primary label classification task, while being non-discriminative when identifying the forget set samples from the validation ones.

Using a similar training procedure as `DANN`, `SURE` fine-tunes the original network features to remove any signature of the forget set examples on the learned representation space while maintaining classification performance.

Let $L_y$ and $L_d$ denote the loss functions for training the label classification module and the domain regressor respectively, defined as

$$L_d(\hat{d}, d) = d \log \frac{1}{\hat{d}} + (1 - d) \log \frac{1}{1 - \hat{d}}, \quad \text{and} \tag{1}$$

$$L_y(\hat{y}, y) = - \sum_k y_k \log \hat{y}_k \tag{2}$$

for arbitrary inputs $(\hat{d}, d)$ and $(\hat{y}, y)$. The network is then trained by performing back-propagation on the sum of the following two objectives capturing the classification and discrimination tasks:

$$\text{(OBJ1)} \qquad \frac{1}{|D_r|} \sum_{(x,y) \in D_r} L_y(G_y(G_f(x; \theta_f); \theta_y), y), \quad \text{and} \tag{3}$$

$$\text{(OBJ2)} \qquad \frac{\lambda}{|D_f^x|} \sum_{x \in D_f^x \cup D_{\text{val}}^x} L_d(G_d(\mathcal{R}(G_f(x; \theta_f)); \theta_d), \mathbb{1}(x \in D_f^x)), \tag{4}$$

where $\mathcal{R}$ represents a so-called gradient reversal layer, such that $\mathcal{R}(x) = x$ on the forward pass and $\mathrm{d}\mathcal{R}(x)/\mathrm{d}x$ is the negative identity matrix on the backward pass. In Appendix C, we describe how this approach defines a saddle-point objective that the domain regressor maximizes and the feature extractor and classification modules minimize. (Note that we assume $|D_f^x| = |D_{\text{val}}^x|$. If not, the sum in Eq. (4) would need to weigh each term in inverse proportion to the respective size of each set.)

The feature-extractor and classification modules, as defined in Section 2.1, are initialized with weights learned during pre-training on $D$. The domain-regressor is initialized randomly.

## 5 EXPERIMENTS

In this section, we present our findings showing that our method not only achieves a superior trade-off between performance and unlearning efficiency compared to existing approaches but also addresses some of the limitations of current unlearning techniques. Specifically, our method mitigates the instability issues seen in previous methods, where results can change significantly with minor adjustments to the unlearning setup-compared to the one used for hyperparameter tuning (Fan et al.,

Table 1: Gap in Top-1 and Top-5 accuracy across all accuracy metrics for the four top performing unlearning methods. For each metric, the best values are highlighted in bold.

| | Top-1 Accuracy | | | | Top-5 Accuracy | | | |
|---|---|---|---|---|---|---|---|---|
| **Metric** | **SURE** | **L1Sparse** | **SCRUB** | **RL** | **SURE** | **L1Sparse** | **SCRUB** | **RL** |
| $\Delta$**RA** | .46±.10 | **.03±.02** | 1.50± .64 | 7.82±2.09 | **.42±.07** | 2.15±.50 | 1.33±.51 | 6.37 ±2.24 |
| $\Delta$**UA** | **1.0±.63** | 3.0±1.10 | 2.±2.53 | 4.2 ±2.56 | **2.52±1.2** | 3.0±1.46 | 3.88±2.57 | 14.64±2.50 |
| $\Delta$**Utility** | **.34±.27** | .62±.26 | 1.26±.30 | .40±.21 | **.66±.16** | 1.48± .14 | 1.02± .36 | .88± .81 |
| $\Delta$**CTA** | **.84±.28** | 2.04±1.57 | 1.68± 1.15 | 2.34±1.56 | 2.41±.96 | 2.53±1.07 | **2.24±1.23** | 3.34±.56 |

2023). In addition, SURE reduces vulnerabilities to basic membership inference attacks in the representation space that are exhibited by some other methods. We assess our method across three distinct unlearning scenarios: *random unlearning*, where forget samples are randomly selected from the entire training set; *partial class unlearning*, where the forget set ($D_f$) is a randomly chosen subset of a specific class; and *class unlearning*, where the forget set ($D_f$) comprises all samples from a particular class, with the primary focus on the partial class unlearning scenario.

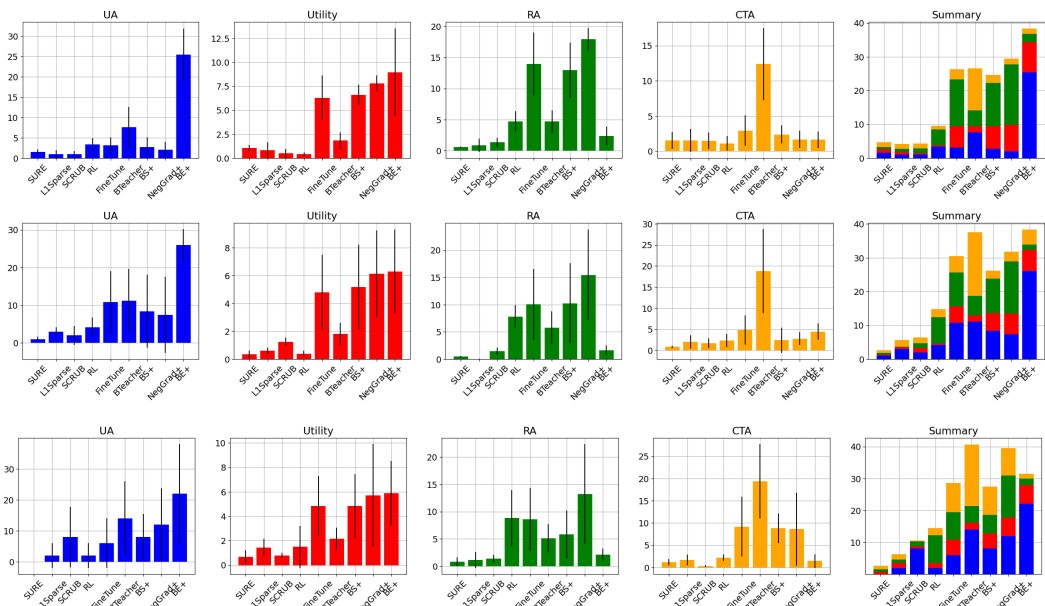

Figure 4: Gap in Top-1 accuracy for forget set sizes of 500 (top), 100 (middle), and 10 (bottom). All three experiments employed the same set of hyperparameters, tuned specifically for the forget set size of 500. A shorter bar (smaller gap) indicates performance is closer to that of the oracle.

## 5.1 EXPERIMENT SETUP

**Stability Experiments**: To assess the stability of unlearning methods, we conduct two sets of experiments. First, we report the average accuracy over the top 5 epochs rather than just the best epoch, to ensure that the results are not merely due to an anomalously good epoch. Second, we evaluate the methods across various forget set sizes.

**Baselines**: We compare our method with several baselines: **Oracle**, which retrains the model from scratch using only $D_r$; **FineTune**, which fine-tunes the original model on $D_r$; **NegGrad**, which uses gradient ascent on $D_f$; **SCRUB** (Kurmanji et al., 2024), a student-teacher framework where unlearning is achieved by minimizing KL divergence between the student and teacher for $D_r$ and maximizing it for $D_f$; **BTeacher** (Chundawat et al., 2023), another student-teacher framework with a pre-trained good teacher and a randomly initialized bad teacher, where the unlearning loss is the sum of KL divergence between the student and the good teacher for $D_r$, and between the student and

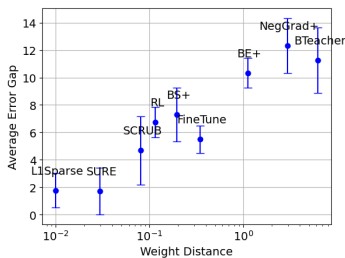

(a) Weight Distance (L2) between oracle and unlearned model vs Average Error Gap for each unlearning method.

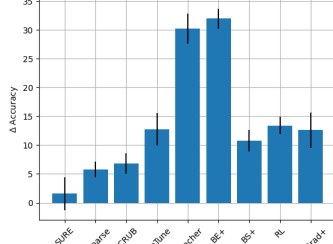

(b) MIARS-I accuracy gap. Attacker is trained to distinguish between forget and test examples.

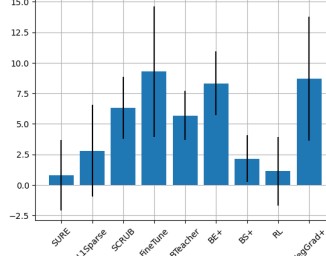

(c) MIARS-II accuracy gap. Attacker is trained to distinguish between training and test examples.

Figure 5: Comparison of the post-unlearning embedding space with that of the oracle using weight comparison (left), MIARS-I attack (middle) and MIARS-II attack (right). The results are plotted across all unlearning methods for the partial class unlearning case with forget size of 10 samples.

the bad teacher for $D_{\mathrm{f}}$; **BE** (Chen et al., 2023) introduces a new class for all forget samples, which is then removed; **BS** (Chen et al., 2023) replaces each forget sample's label with that of a similar image from a different class, based on gradient space similarity; **L1Sparse** (Liu et al., 2024) applies L1 regularization to the objective function to promote sparsity and facilitate unlearning and finally **RL** replaces forget sample labels with random ones.

It is important to note that we faced difficulties achieving good results with **NegGrad**, **BE**, and **BS** due to instability during unlearning. To address this, inspired by Kurmanji et al. (2024), we enhanced these methods by incorporating an additional term into the objective function: cross-entropy loss trained on the $D_{\mathrm{r}}$. We refer to these enhanced methods as **NegGrad+**, **BE+**, and **BS+** respectively.

**Implementation Details**: Extensive hyperparameter tuning was conducted independently for each method. The primary hyperparameters optimized were the learning rate and $\alpha$, where $\alpha$ controls the weight of the retain set loss in methods with a weighted loss function that combines losses for the retain and forget sets. Additional hyperparameters, including batch size for the forget set and learning rate scheduling, were also tuned. Optimal hyperparameters for each model were determined through Bayesian optimization to achieve the best unlearning to utility trade-off. The same original model was used across all unlearning methods, trained for 150 epochs with a learning rate of 0.01 and a weight decay of 0.0005, without data augmentation, with the learning rate reduced by an order of magnitude at epochs 80 and 150. For the random unlearning method, hyperparameter tuning was carried out with a forget set comprising 5,000 samples. In contrast, for (partial) class unlearning, the forget set was considered to contain 500 samples. In the (partial) class unlearning experiments, we initially selected one class at random (class 5) and conducted all experiments for that particular class.

**Additional Experiments**: To gain further insight into our method's effectiveness, we conduct additional experiments involving different datasets, architectures, benchmarks, and analyses, which are not included here due to space constraints. These results are provided in the appendices. More specifically, we evaluate our method on two additional datasets: CINIC-10 (Darlow et al., 2018), an extension of CIFAR-10 augmented with samples from ImageNet, and Tiny ImageNet (Le & Yang, 2015) in Appendix D. We also test our method on ViT-Small, an example of a larger model architecture, trained on CIFAR-10, as detailed in Appendix D. Additionally, we investigate the effectiveness of our method on a non-privacy-related benchmark, Removing Confusion Kurmanji et al. (2024), in Appendix E. Finally, we conduct a sensitivity analysis to evaluate the method's robustness to slight changes in hyperparameters, as presented in Appendix F.

## 5.2 EXPERIMENT RESULTS

In this section, we examine the effectiveness of our method against both oracle and multiple baselines across various unlearning benchmarks.

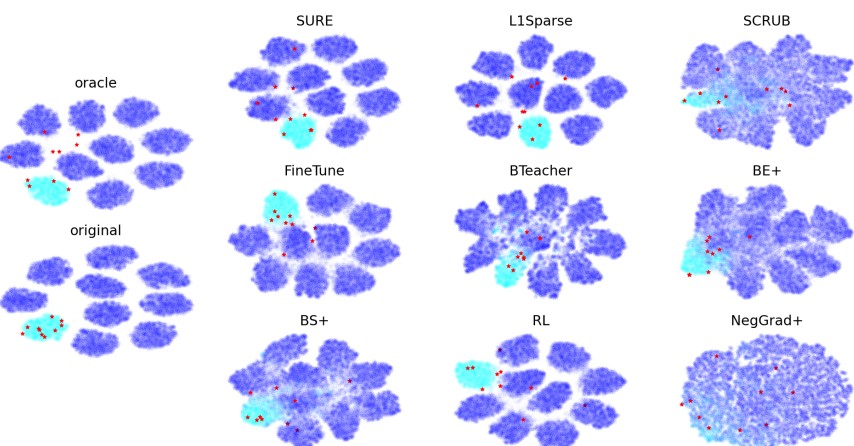

Figure 6: t-SNE visualization of the post-unlearning embedding space for the entire CIFAR-10 training set across all unlearned models. The first column compares these embeddings against the original (top) and oracle models (bottom). Cyan represents retain samples from the forget class, blue indicates other retain samples, and red denotes the 10 forget samples from the same class.

**Result reporting:** For each experiment, results are presented for the *best* epoch, defined as the one for which the utility versus privacy trade-off is optimized. Since most baselines, except for SCRUB, did not provide an explicit selection process, we applied our own method to determine the epoch for reporting results. For SCRUB, the *best* epoch is selected according to the criteria outlined in the original paper (corresponding to the SCRUB-R version of the algorithm). Additionally, unless stated otherwise, the figures and tables in this section do not present the performance of the unlearned models, but rather the gap between their performance and that of the oracle for a particular benchmark. The gap is calculated as the absolute value of their differences; thus, a smaller value indicates better performance of the method. The difference is averaged over five separate runs, with the original model remaining constant while the randomly selected forget set varies. Error bars represent the variance across these five runs.

**Accuracy and MIA**: Fig. 3 presents a summary of the results for three key sets of benchmarks: Accuracy metrics, MIAs and RRT, across all three unlearning scenarios and methods (see Appendix B for more detailed plots). Overall, our method outperforms all other unlearning methods in most benchmarks across these scenarios. In the random unlearning scenario, the L1Sparse method closely follows ours. While SURE outperforms L1Sparse overall all but one of the benchmarks, L1Sparse achieves better RRT. This discrepancy is primarily due to the additional module in our approach—the domain regressor—which tends to slow down performance compared to the others. Regarding MIA attacks, our results indicate that these attacks are largely ineffective in this scenario, with all methods, including the oracle, yielding similar outcomes. In contrast, we observed that these attacks are more effective in class unlearning scenarios. Notably, our method excels in these scenarios, outperforming all other methods across all accuracy benchmarks and both MIA attacks while maintaining competitive RRT.

**Canary Unlearning**: Fig. 2 presents the accuracy results from the canary unlearning experiment. SURE outperforms all other methods across all accuracy measures. It effectively unlearns both the IN and OOD samples while maintaining strong generalizability (Utility) and fidelity (RA).

**Representations and MIARS**: The primary motivation for our method is the hypothesis that a correlation exists between the similarity of the post-unlearning model's representation to that of the oracle and the effectiveness of the unlearning process. To investigate this hypothesis, we plot the *Average Error Gap* against the L2 distance between the penultimate layer's weights of the unlearned model and the oracle, as shown in Fig. 5a. The *Average Error Gap* is defined as the average absolute value of the difference between the unlearned model and the oracle across five key unlearning benchmarks: UA, Utility, RA, CTA, and MIARS-I. The results clearly indicate a correlation be-

tween the alignment of the unlearned representation with the oracle and the success of unlearning. In addition, our method has the lowest *Average Error Gap* and second smallest weight distance, After L1Sparse. The t-SNE visualization of the embedding space for all unlearned models, shown in Fig. 6, further suggests a correlation. Notably, the most effective unlearning methods, SURE followed by L1Sparse, exhibit representations that closely resemble the oracle.

In addition to subpar unlearning performance, discrepancies between the unlearned model and the oracle in the representation space can render the model vulnerable to simple MIAs. The results are shown in Fig. 5b and 5c. As with previous plots, the y-axis indicates the difference in MIA accuracy between the unlearned model and the oracle. The model unlearned with SURE exhibit is most successful in defending both attacks. Also, overall models whose representation is closer to oracle (measured by weight distance) are less susceptible to attacks.

**Stability**: For an unlearning method to be effective in real-world applications, it must demonstrate stability. We assess stability by conducting two sets of experiments. In the first set of experiments, we report the top-5 accuracy, calculated as the average accuracy over the top 5 epochs for the four highest-performing unlearning methods, in Table.1. We compare these results to the top-1 accuracy, which reflects the performance from the *best* epoch. Our findings indicate that SURE outperforms the other three methods overall and across most individual metrics.

In the second experiment, we perform unlearning for different forget sizes (100 and 10), other than the one used for hyperparameter tuning (500). This experiment assesses how sensitive each unlearning method is to slight changes in the setup for which it is optimized. The results are presented in Fig. 4. While L1Sparse and SCRUB both show performance comparable to SURE for the forget set size used for hyperparameter tuning (top plot), the gap between their performance and that of SURE widens as the difference between the forget set used for unlearning and the original forget set grows (middle and bottom plots).

# 6 CONCLUSION AND FUTURE WORK

In this work, we propose SURE, an approximate unlearning method that focuses on unlearning in the representation space rather than solely in the output space. This approach is motivated by our observation that achieving similarity with the oracle in the representation space leads to more successful unlearning. Through extensive experiments, we demonstrate that our method consistently yields superior or comparable results compared to existing techniques across a range of scenarios and for a variety of unlearning benchmarks, including multiple accuracy metrics and two MIAs. Furthermore, we empirically validate that the post-unlearning representation of the unlearned model closely resemble that of the oracle. Finally, our approach exhibits greater stability and reduced susceptibility to white-box attacks in the representation space, enhancing overall model security.

One observed limitation of our method is the need for more unlearning updates compared to other techniques, especially in the random unlearning scenario where forget samples are randomly selected from any class. A potential solution could be to employ a class-conditional discriminator instead of a generic one, allowing the discriminator to focus on differentiating between class-specific forget and validation samples. While promising, such an approach would come at a cost of training and storing multiple discriminators.

ACKNOWLEDGEMENT

We thank Mila – Quebec AI Institute and Google DeepMind for providing the computational resources that supported this work. We also sincerely appreciate Devin Kwok for his valuable input and for providing the implementation code, both of which were instrumental for our analyses in the representation space. We also thank Katja Filippova for providing feedback on the drafts of this work. Daniel M. Roy is supported by the funding through NSERC Discovery Grant and Canada CIFAR AI Chair at the Vector Institute.

REPRODUCIBILITY

To ensure reproducibility, we provide a comprehensive outline of our experimental setup in Section 5.1, including details on hyperparameters, hyperparameter tuning strategies, datasets, and algorithms. The precise hyperparameter settings are further documented in the Appendix G. Our stability experiments, as detailed in Section 5.2, significantly enhance reproducibility and reveal inherent instability in some previously proposed methods. Furthermore, we address a critical and often overlooked aspect of unlearning algorithms in Section 5.2 by carefully outlining our stopping criteria selection process. Finally, all average numerical values in our experimental results are presented with error bars computed over 5 independent runs.

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

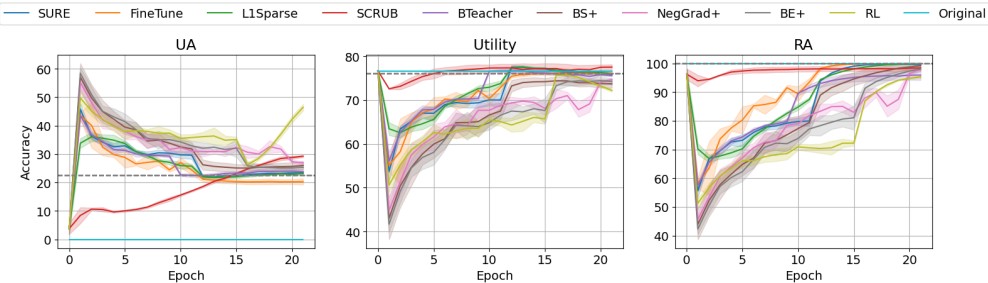

Figure 7: Top-1 accuracy for the three accuracy metrics, as indicated in the title of each plot, during unlearning across all unlearned models for the random unlearning case. In contrast to the results section, the values shown here represent the actual accuracy rather than the gap. The gray dashed line in each plot illustrates the performance of the oracle.

# A    RELATED WORK

Unlearning was first introduced in (Cao & Yang, 2015) who proposed a forgetting algorithm for statistical query learning. Nowadays, a plethora of unlearning methods have been proposed that attempt to selectively remove the influence of the forget set from the trained model. Golatkar et al. (2020a) introduced an information-theoretic method that uses a Newton step and adds noise to erase the forget set. Golatkar et al. (2020b) proposed NTK forgetting, employing a first-order Taylor expansion to estimate the weights that would have been obtained if training without the forget set. These methods work well under assumptions of stability of SGD. Izzo et al. (2021); Koh & Liang (2017) proposed unlearning algorithms based on influence functions Cook & Weisberg (1982), drawing connections to $(\epsilon, \delta)$-*forgetting* (Guo et al., 2019). Goel et al. (2022a) introduced two approximate unlearning methods: Catastrophic Forgetting-k(CF-K) and Exact Unlearning-k (EU-k). In both methods, the first $k$ layers of the pre-trained model are frozen. In CF-K, the remaining layers are fine-tuned on $D_r$ while in EU-k forgetting, they are trained from scratch on $D_r$. (Chundawat et al., 2023; Kurmanji et al., 2024) proposed teacher-student formulations, where the teacher is the original model and the student is initialized from the teacher and, through specially-crafted distillation procedures (where the teacher is frozen and the student is updated), turns into the unlearned model. In (Chundawat et al., 2023), the student distills retain set knowledge from the teacher, while being encouraged to agree with another "incompetent" teacher on the forget set, whereas SCRUB (Chundawat et al., 2023) considers only one teacher, and encourages the student to agree with it on the retain set while disagreeing with it on the forget set. Chen et al. ((Chen et al., 2023)) adopted a unique strategy for class unlearning by adjusting the decision boundary of the original model to mimic the oracle's decision-making. They introduced two boundary shift methods: Boundary Expanding, which creates a new class for all forget samples that will later be removed, and Boundary Shrinking, which assigns forget samples labels from similar images in different classes. Liu et al. (2024) proposed L1Sparse, adding an L1-penalty into the fine-tuning baseline, after conducting an investigation that shows that sparsity makes unlearning easier. Fan et al. (2023) proposed a localized unlearning approach that targets only a subset of parameters that are deemed-to-be the most critical for the forget set.

# B    UNLEARNING RESULTS FOR CIFAR-10

Fig. 7 and Fig. 10 illustrate the accuracy metrics (UA, RA, and Utility, and in the case of class unlearning, CTA) during the unlearning process across different methods for random and partial class unlearning scenarios respectively. The gray dashed line in each plot represents the oracle performance.

The label *Original* denotes the original model which serves as the starting point for all unlearning methods. To evaluate the effectiveness of each unlearning method we compare the performance of the unlearned model with that of the oracle on all accuracy metrics. For each unlearning method, the comparison is made at the *best* epoch. The absolute value of the difference between the unlearned model and the oracle model for each method is displayed in Fig. 8 and Fig. 11. In the random unlearning scenario, the forget set consists of 1,000 samples randomly selected from the entire

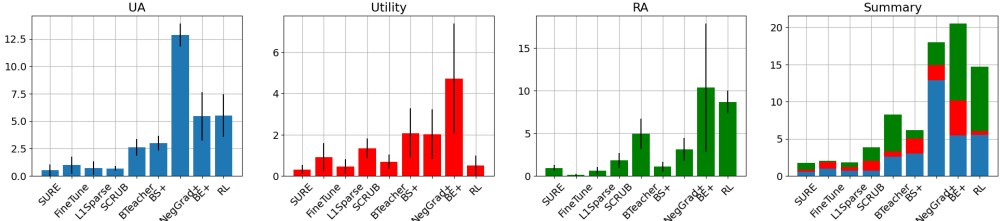

Figure 8: Gap in Top-1 accuracy for various accuracy metrics, as indicated in the title of each plot, for the random unlearning scenario with a forget size of 1000, across all unlearned models. Results are reported for the *best* epoch.

training set, while in the partial unlearning scenario, it consists of 100 samples randomly chosen from class 5.

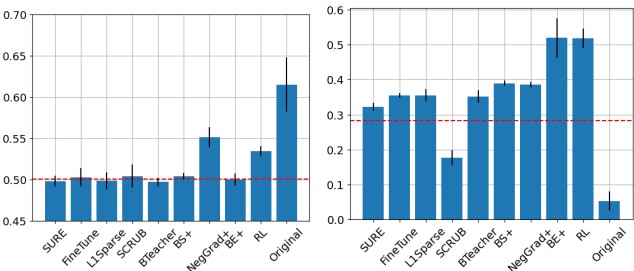

Figure 9: MIA-I (left) and MIA-II (right) in output space across all unlearned models for random unlearning. Red line represents the performance of the oracle.

The results for MIA-I and MIA-II in the random and partial unlearning cases are shown in Fig. 9 and Fig. 12, respectively. The red line indicates the attack accuracy on the oracle, serving as a reference point for all unlearning methods. MIA-I does not appear to be effective in the random unlearning scenario, as nearly all unlearning methods seem equally successful at defending against it. However, this is not the case for the partial class unlearning scenario, where our method's success rate is very close to that of the oracle. For MIA-II, our method outperforms the others, showing an overall success rate closest to the oracle's for both random and partial unlearning scenarios, although the results for the random scenario are quite similar across all methods.

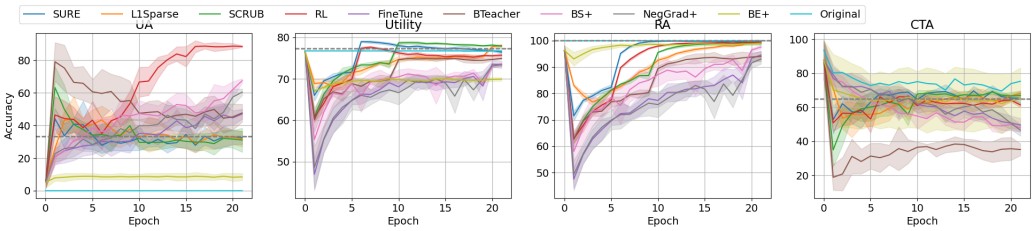

Figure 10: Top-1 accuracy across all unlearned models for the partial class unlearning case with the forget size of 100. Each panel presents a different accuracy metric, as indicated in the title. The gray dashed line in each plot illustrates the performance of the oracle.

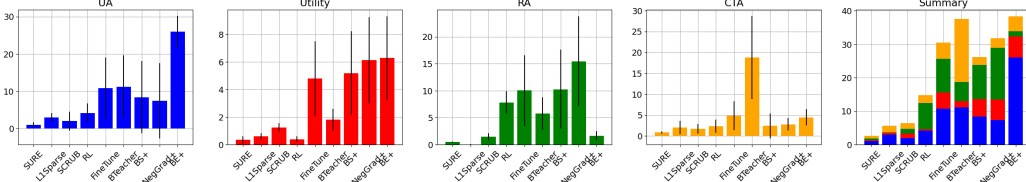

Figure 11: Gap in Top-1 accuracy for various accuracy metrics for the partial unlearning case across all unlearned models.

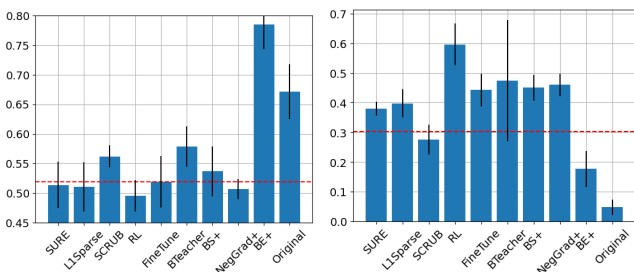

Figure 12: MIA-I (left) and MIA-II (right) results for the partial class unlearning scenario. Red line represents the performance of the oracle.

## C  GRADIENT UPDATES

The purpose of this section is to describe the mathematical optimization problem underlying training. We will see that it is implemented succinctly with a gradient reversal layer (GRL), which is how we describe it in the main paper (Eqs. (3) and (4)). Readers may find one of these two equivalent descriptions more intuitive.

Define

$$E(\theta_f, \theta_y, \theta_d) = \frac{1}{|D_r|} \sum_{(x,y) \in D_r} L_y(G_y(G_f(x; \theta_f); \theta_y), y)$$
$$- \frac{\lambda}{|D_f^x|} \sum_{x \in D_f^x \cup D_{val}^x} L_d(G_d(G_f(x; \theta_f)); \theta_d), \mathbb{1}(x \in D_f^x)).$$

This objective is the objective underlying domain adversarial neural networks (Ganin et al., 2016). The first term captures the training error on the classification task and depends on the feature extractor and classification modules. The second term captures the error on the discrimination task, and depends on the feature extractor and domain-regressor modules.

We optimize this objective for training in SURE. In particular, the goal of domain adversarial training is to find a saddle point, where the optimal parameters for each module satisfy

$$(\hat{\theta}_f, \hat{\theta}_y) = \underset{\theta_f, \theta_y}{\arg\min} \, E(\theta_f, \theta_y, \hat{\theta}_d), \tag{5}$$

$$\hat{\theta}_d = \underset{\theta_d}{\arg\max} \, E(\hat{\theta}_f, \hat{\theta}_y, \theta_d). \tag{6}$$

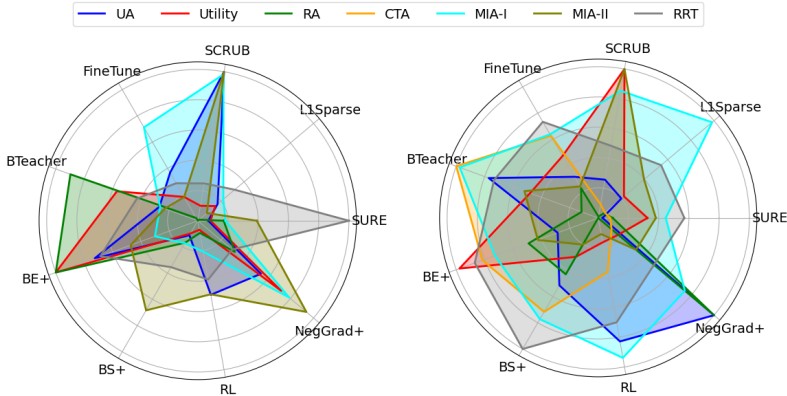

Figure 13: Results for the random and partial class unlearning scenarios on CINIC-10. Each benchmark plot illustrates the absolute difference between the unlearned model and the oracle. For random unlearning, the forget set contains 1,000 samples, while for partial unlearning, it consists of 100 samples. The results are averaged over five distinct runs, each with a unique forget set. To enhance visual clarity, the values for each metric have been normalized to their maximum value.

Following Ganin et al. (2016), we use the following gradient updates to seek a saddle-point:

$$\theta_f \leftarrow \theta_f - \mu\Big(\frac{\partial L_y}{\partial \theta_f} - \lambda\frac{\partial L_d}{\partial \theta_f}\Big), \tag{7}$$

$$\theta_y \leftarrow \theta_y - \mu\Big(\frac{\partial L_y}{\partial \theta_y}\Big), \tag{8}$$

$$\theta_d \leftarrow \theta_d - \mu\lambda\Big(\frac{\partial L_d}{\partial \theta_d}\Big), \tag{9}$$

where $\mu$ is the learning rate. As discussed in (Ganin et al., 2016), these updates closely resemble SGD updates for a simple feed-forward deep model objective function, with one key difference: the gradients from the domain regressor are subtracted instead of added. This adjustment ensures that, while the regressor's task is to distinguish between forget and validation samples, the feature extractor aims to deceive the regressor. As a result, the feature extractor learns a set of indiscriminative features that cannot differentiate between the forget samples and previously unseen samples.

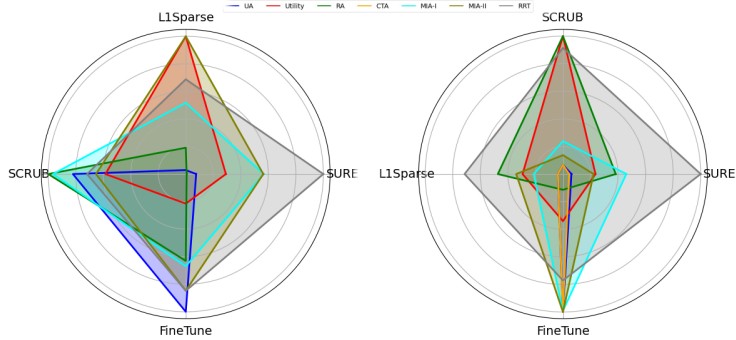

Figure 14: Results for the random and partial class unlearning scenarios on the Tiny ImageNet dataset. For random unlearning, the forget set contains 1,000 samples, while for partial class unlearning, it contains 400 samples. Results are averaged over five runs, each with a unique forget set.

As noted by Ganin et al. (2016), this modification to the usual update rules is cumbersome to implement naively in most deep learning libraries. To address this, the authors introduced a special "gradient reversal layer" (GRL), which has no associated parameters. During forward propagation,

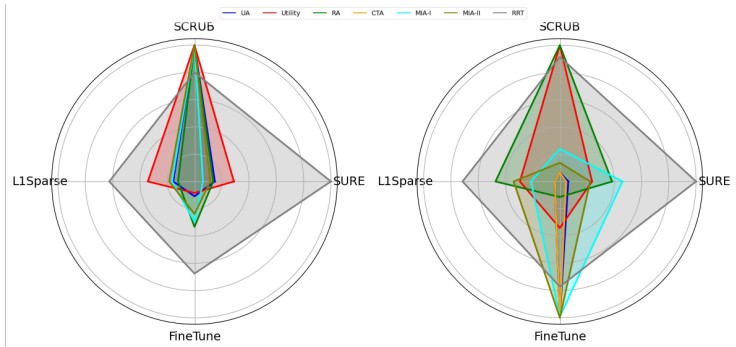

Figure 15: Results for the random and partial class unlearning scenarios using a Small-ViT trained on CIFAR-10. For random unlearning, the forget set contains 1,000 samples, while for partial class unlearning, it contains 10 samples. Results are averaged over five runs, each with a unique forget set.

the GRL acts as an identity transformation. During backpropagation, however, it takes the gradient from the subsequent layer and reverses its sign (i.e., multiplies it by the negative identity) before passing it to the preceding layer. Because the computation involving the domain regressor weights comes after the GRL, gradient descent *minimizes* the discrimination task objective in the domain regressor weights. In contrast, the computation involving the feature extractor weights come before the GRL, and so gradient descent *maximizes* the discrimination task objective in the feature extractor weights.

In conclusion, if we perform backpropagation of the computation representing *sum* of Eq. (3) and Eq. (4), the latter of which includes the GRL, we obtain updates identical to Eq. (7). This objective with a GRL can be optimized using standard auto-differentiation tools.

| Method | Retain Err | Forget Err | Test Err | IC Retain | FGT Retain | IC Test | FGT Test |
|---|---|---|---|---|---|---|---|
| SURE | $0.08 \pm 0.11$ | $12.25 \pm 4.57$ | $22.75 \pm 0.31$ | $0.0 \pm 0.0$ | $0.4 \pm 0.49$ | $0.16 \pm 0.01$ | $29.4 \pm 3.88$ |
| SCRUB | $0.32 \pm 0.3$ | $10.5 \pm 2.03$ | $25.36 \pm 1.01$ | $0.0 \pm 0.0$ | $4.0 \pm 4.86$ | $0.17 \pm 0.01$ | $35.6 \pm 6.56$ |
| L1Sparse | $0.08 \pm 0.11$ | $12.25 \pm 3.1$ | $22.18 \pm 0.53$ | $0.0 \pm 0.0$ | $0.2 \pm 0.4$ | $0.15 \pm 0.01$ | $27.0 \pm 3.63$ |
| FineTune | $2.77 \pm 0.73$ | $29.75 \pm 5.33$ | $26.89 \pm 0.46$ | $0.07 \pm 0.01$ | $2.6 \pm 1.85$ | $0.32 \pm 0.01$ | $17.4 \pm 3.61$ |
| BE+ | $0.0 \pm 0.0$ | $100.0 \pm 0.0$ | $28.61 \pm 0.58$ | $0.58 \pm 0.12$ | $25.0 \pm 13.61$ | $0.64 \pm 0.08$ | $10.0 \pm 5.4$ |
| BS+ | $2.13 \pm 1.98$ | $85.75 \pm 3.92$ | $34.14 \pm 1.66$ | $0.07 \pm 0.04$ | $7.8 \pm 15.6$ | $0.45 \pm 0.02$ | $13.0 \pm 3.22$ |
| RL | $13.65 \pm 1.25$ | $19.75 \pm 3.1$ | $26.53 \pm 0.48$ | $0.07 \pm 0.01$ | $60.4 \pm 14.36$ | $0.17 \pm 0.01$ | $38.4 \pm 6.44$ |
| NegGrad+ | $0.0 \pm 0.0$ | $100.0 \pm 0.0$ | $28.6 \pm 0.58$ | $0.5 \pm 0.0$ | $3960.0 \pm 0.0$ | $0.5 \pm 0.0$ | $1000.0 \pm 0.0$ |
| BTeacher | $1.17 \pm 0.29$ | $8.75 \pm 3.45$ | $27.67 \pm 0.57$ | $0.05 \pm 0.01$ | $2.0 \pm 2.61$ | $0.35 \pm 0.01$ | $11.6 \pm 3.2$ |

Table 2: Performance comparison of multiple unlearning methods on the RC task across various metrics. Results are averaged over 5 runs.

# D  MORE UNLEARNING RESULTS

In this section, we evaluate the performance of our unlearning method across additional datasets and architectures. Specifically, we extend our analysis to CINIC-10, Tiny ImageNet, and ViT-Small, comparing our method against other unlearning approaches across various benchmarks.

In Fig. 13, we compare the performance of our method on the CINIC-10 dataset against all other baselines across various unlearning metrics, similar to Fig. 3. The figure displays the absolute difference between the unlearned model and the oracle for all accuracy metrics (UA, RA, Utility, and CTA), as well as for RRT and MIAs. We observe that SURE outperforms most benchmarks in both scenarios. Consistent with the results from CIFAR-10, L1Sparse ranks as the second-best performing method, occasionally surpassing SURE on a few metrics.

In addition, we evaluate the performance of our method on the Tiny ImageNet dataset, comparing it against three other top-performing methods identified in previous experiments: L1Sparse, FineTune, and SCRUB. The results, presented in Fig. 14, show that our method outperforms all others across all benchmarks except for RRT, consistent with our findings on CIFAR-10 and CINIC-10. For random

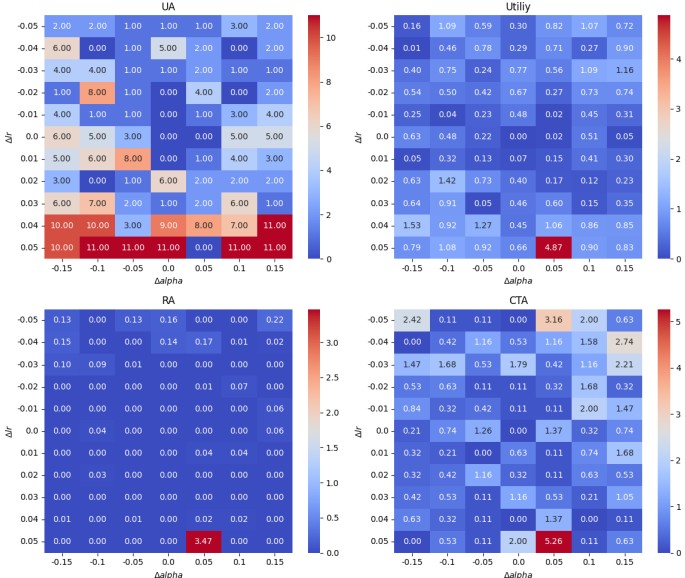

Figure 16: Heatmaps illustrating the sensitivity analysis of SURE's performance with respect to the two key hyperparameters: $lr$ and $\alpha$, across the four main accuracy benchmarks: UA (top-left), Utility (top-right), RA (bottom-left), and VTA (bottom-right). Each heatmap shows performance variations relative to the best-tuned hyperparameter settings.

unlearning, we were unable to identify a set of hyperparameters for FineTune and SCRUB that achieved successful unlearning. Both methods fail to achieve the desired unlearning-utility tradeoff, as they completely unlearn all information. In the case of class unlearning, FineTune and SCRUB are more competitive; however, the primary competition remains between our method (SURE) and L1Sparse, with SURE outperforming L1Sparse in nearly all benchmarks.

To explore the effectiveness of our method on larger models, we tested its performance using ViT-Small across both random and class unlearning scenarios. Results are presented in Fig. 15. For random unlearning, our method demonstrates a slight edge over L1Sparse. We were unable to find hyperparameters for SCRUB that enabled effective unlearning, as models using this method failed to achieve the desired unlearning-utility tradeoff. For class unlearning, the results are more balanced: our method outperforms L1Sparse on 4 out of 7 benchmarks, while L1Sparse outperforms on the remaining 3. These findings highlight the robustness of our approach, while also demonstrating that pruning-based methods like L1Sparse can remain competitive in specific settings.

# E  UNLEARNING RESULTS FOR THE RESOLVING CONFUSION TASK

In this section, we apply our unlearning method to the Resolving Confusion (RC) task introduced in Kurmanji et al. (2024). This task serves as an example of a non-privacy-related application where unlearning is beneficial for resolving confusion between classes caused by a portion of the model's original training set being mislabeled.

We evaluated the performance of `SURE` on the RC task using a ResNet-18 model trained on the CIFAR-10 dataset. We compared our method against all eight competing unlearning methods introduced in the paper. In our implementation, the forget set consists of a randomly selected subset (1%) of training samples from classes 0 and 1, which are intentionally mislabeled as the opposite class. Given that all samples in the forget set are mislabeled, the objective of unlearning in this context is to resolve the induced confusion. An unlearning method is considered successful if, at the end of the unlearning process, the forget samples are correctly relabeled while maintaining the model's performance on the test and retain sets.

To evaluate the performance of our unlearning method, we report the forget error (Forget Err) alongside the test (Test Err) and retain errors (Retain Err). An effective method is expected to exhibit low

error values across all three metrics. In addition to these, and in line with the original paper Kurmanji et al. (2024), we also report the Interclass Confusion Error (IC-ERR) and FGT-ERR. Unlike the previously introduced errors, which count all types of incorrect predictions, IC-ERR specifically measures errors involving the confused classes (in our setup, class 0 and class 1). It counts instances where a sample from class 0 is predicted to belong to any other class, and vice versa for class 1. FGT-ERR, on the other hand, is even more specific, as it counts only cases where a sample from class 0 is mislabeled as class 1, or a sample from class 1 is mislabeled as class 0. IC-ERR and FGT-ERR are reported for both retain and test sets. Our results presented in Table 2 demonstrate that SURE is the top-performing approach, alongside L1Sparse, yielding similar results for many metrics.

## F    SENSITIVITY ANALYSIS OF HYPERPARAMETERS

To assess the sensitivity of our method's performance to variations in its hyperparameter values, we conducted a sensitivity analysis focusing on the two most critical hyperparameters: $\alpha$ and the learning rate ($lr$). This analysis was carried out for the class unlearning scenario using a ResNet-18 model trained on the CIFAR-10 dataset, with a forget set consisting of 100 samples. Using a grid search approach, we varied each parameter's value within a range of -20% to +20% around its optimal value and analyzed the resulting changes across multiple unlearning benchmarks.

The observed performance variations are illustrated in Fig. 16. The figure presents four heatmaps, each providing a sensitivity analysis for a specific accuracy benchmark as a function of these hyperparameters. These heatmaps highlight the differences in performance relative to the model trained with the best hyperparameter configuration from tuning. The visualizations reveal the sensitivity to each parameter and their interactions, showing that, compared to other methods (see Section 4 and Figure 2 in Fan et al. (2023)), SURE demonstrates low sensitivity to hyperparameter variations. For comparison, Fan et al. (2023) reports fluctuations of approximately 40 points in the average performance gap between the unlearned model and the oracle for the Influence Unlearning method. While there is always room for improvement, our results suggest that sensitivity to hyperparameters is not a significant limitation of SURE.

## G    UNLEARNING HYPERPARAMETERS

The optimized values for all hyperparameters across unlearning methods in both random and partial class unlearning scenarios are presented in Table 3 and Table 4, respectively. The main hyperparameters we tuned in this work are the learning rate, $lr$, $\alpha$ (retain set loss weight in unlearning methods with a weighted objective function), forget set batch size, and the learning rate scheduling, defined by the epoch at which the learning rate drops.

| Params | SURE | L1Sparse | SCRUB | FineTune | Bteach | BE+ | BS+ | RL | NegGrad+ |
|---|---|---|---|---|---|---|---|---|---|
| $rl$ | .045 | .0045 | .0008 | .0043 | .035 | .0092 | .0079 | .0671 | .0072 |
| $\alpha$ | .807 | .00044 | .997 | - | .954 | .982 | .998 | - | .991 |
| fbatch size | 256 | 32 | 256 | 256 | 8 | 128 | 128 | 32 | 8 |
| $lr$ decay | 10 | 10 | 18 | 8 | 8 | 14 | 10 | 14 | 18 |

Table 3: Optimized hyperparameters for the random unlearning scenario: learning rate ($lr$), $\alpha$, forget set batch size (fbatch size), and the epoch for learning rate decay ($lr$ decay).

| Params | SURE | L1Sparse | SCRUB | FineTune | Bteach | BS+ | BE+ | RL | NegGrad+ |
|---|---|---|---|---|---|---|---|---|---|
| $lr$ | .0372 | .0038 | .001 | .0025 | .0320 | .0047 | .0058 | .0014 | .0694 |
| $\alpha$ | .4241 | .00024 | .6089 | - | .3453 | .9841 | .4639 | - | .9981 |
| fbatch size | 128 | 256 | 32 | 32 | 64 | 8 | 128 | 64 | 256 |
| $lr$ decay | 4 | 18 | 6 | 18 | 8 | 12 | 4 | 10 | 18 |

Table 4: Optimized hyperparameters for the (partial) class unlearning scenario: learning rate ($lr$), $\alpha$, forget set batch size (fbatch size), and the epoch for learning rate decay ($lr$ decay).

