# OpenReview forum: "Selective Unlearning via Representation Erasure Using Domain Adversarial Training"
_ICLR.cc/2025/Conference — ICLR 2025 Poster_

### Official Review · Reviewer_5UH3 · 2024-11-01

**Soundness:** 4
**Presentation:** 3
**Contribution:** 3
**Rating:** 6
**Confidence:** 4

**Summary:**

The paper proposes SURE (Selective Unlearning via Representation Erasure), a novel approach for targeted unlearning in neural networks, based on representation erasure rather than model output manipulation. Using concepts from Domain-Adversarial Training of Neural Networks (DANN), SURE aims to align the representation space of a model post-unlearning with that of an oracle model. This method focuses on defending against more robust attacks, particularly membership inference attacks (MIAs) in the representation space, improving the stability of unlearning while preserving model utility.

**Strengths:**

- The paper introduces a unique perspective by targeting the representation space instead of the output space, addressing a critical gap where unlearned models might retain information in their intermediate representations. This angle has the potential to advance understanding and methodology in the field of machine unlearning.
- The authors validate SURE’s effectiveness using extensive experiments across several unlearning scenarios, including random, partial, and full class unlearning. The results consistently demonstrate that SURE achieves a favorable trade-off between performance and unlearning efficiency.
- By specifically aiming to improve defenses against membership inference attacks within the representation space, SURE contributes to increasing the security and robustness of unlearned models.
- The technical approach, particularly the adaptation of DANN with gradient reversal and domain-adversarial training, is well-articulated.

**Weaknesses:**

- Although SURE demonstrates stability across different forget set sizes, its reliance on careful hyperparameter tuning (with Bayesian optimization) may hinder ease of implementation for practitioners who cannot afford extensive hyperparameter searches.
- SURE appears to introduce additional computational complexity. While this approach enhances unlearning quality, it may limit SURE's scalability, especially for larger models or datasets where computational efficiency is crucial.
- Minor Comment on Notation Consistency: The authors should maintain consistent notations across the paper to improve readability. For example, in Figure 1, the notation for different modules should be consistent with those used in the main text ($G_f(., \theta_f)$, $G_y(., \theta_y)$, and $G_d(., \theta_d)$). This would help readers follow the mathematical formulations and understand the relationships among components more easily.

**Questions:**

1. Could the authors provide a breakdown of the computational cost of SURE compared to the baselines?
2. How sensitive is SURE’s performance to the choice of hyperparameters, especially those determined by Bayesian optimization? If SURE requires highly tuned hyperparameters, exploring ways to reduce this dependency could make it more accessible for practical applications.

---

> ### Author Response · Authors · 2024-11-28
> **rebuttal_reviewer_5UH3**
>
> Thank you for your review. We address the two key concerns (sensitivity and computational complexity). In short: no, SURE is not sensitive and, while it is not the fastest method, it is in the ballpark. We expect you'll find our arguments convincing and we hope that you'll step up to champion our paper, in order to see that it is published. So far, across all three reviews, there has not been a concern that we have not thoroughly addressed. Best paper? Maybe. Or maybe not. We leave this difficult choice to you.
>
> **Weakness? "SURE's... reliance on careful hyperparameter tuning (with Bayesian optimization) may hinder ease of implementation."** We'd like to push back on the very idea that SURE is sensitive in any unusual way. We don't think SURE is sensitive and we ran a sensitivity analysis below which demonstrates empirically that SURE is no more sensitive than other algorithms like it. Perhaps our use of BayesOpt to do HPT suggested sensitivity was a problem? In our opinion, the use of rigorous methods for HPT is simply good empirical ML hygiene. While it is not common, it should be, as it allows you, the reader and reviewer, to rest assured that we were not cherry picking HPTs to make the comparisons look better. (Another reason for BayesOpt: not all methods we looked at published their HPs!!)
>
> Regardless, we conducted a sensitivity analysis focusing on the two most important hyperparameters of our method: alpha (α) and lr. This is done for the class unlearning scenario for a ResNet-18 trained on CIFAR-10 when the forget set consists of 100 samples. Using a grid search approach, we varied each parameter's value within a range of -20% to +20% around its optimal value and analyzed the resulting changes for multiple unlearning benchmarks. The observed performance changes are illustrated in Appendix E in the paper.  Fig.16 . In short, SURE is no more sensitive.
>
> This plot presents four heatmaps that provide a sensitivity analysis for each accuracy benchmark, showing how the performance of the unlearned model varies with different learning rate (lr) and α values. Each heatmap represents a specific accuracy benchmark as a function of these hyperparameters and highlights the difference in performance relative to the model trained with the best hyperparameter set from HPT. These visualizations reveal the sensitivity to each parameter and their interactions, demonstrating that, compared to other methods (see Section 4 and Figure 2 in [1]), SURE does not exhibit high sensitivity to hyperparameter values. For comparison, [1] reports approximately 40 points of fluctuation in the average gap between the unlearned model and the oracle for the Influence Unlearning method. There's always room for improvement, but sensitivity is not a real weakness of SURE.
>
> [1] Fan C, Liu J, Zhang Y, Wong E, Wei D, Liu S. Salun: Empowering machine unlearning via gradient-based weight saliency in both image classification and generation. arXiv:2310.12508.
>
>
> **Can you comment on the computational complexity of SURE?**  We calculate runtime (RT) by averaging the time per epoch for each method. For unlearning methods, we multiply this average by 20 (number of unlearning epochs), and for training from scratch, by 150 (number of training epochs). The relative runtime (RRT) is then the ratio of the unlearning runtime to the training runtime. As an example, the RT and RRT values for the top four performing unlearning methods in the case of partial unlearning on Tiny-ImageNet are reported below:
>
>
> | Metric        | SURE      | SCRUB      | L1Sparse      | FineTune     | Oracle    |
> |:--------------|:--------- |:-----------|:--------------|:-------------|:----------|
> | RT (sec)      | 29.32     | 20.16      | 21.06         | 24.86        | 20.83     |
> | RRT           | .1877     | .1290      | .1348         | .1591        | N/A       |
>
>
>
> In summary, SURE has more overhead at the moment, but it also represents a very different type of algorithm, and one which has not received the optimization attention (e.g., with respect to caching, throughput, etc.) that usual training has received (and that more standard approaches might benefit from). We believe the method is promising enough to rally other researchers to join the effort to improve it, including optimizing its run time.

---

> > ### Author Response · Authors · 2024-12-02
> > **followup-reviewer_5UH3**
> >
> > Dear Reviewer,
> >
> > We posted our responses to your comments a few days ago and wanted to kindly follow up. Have our replies addressed your concerns? Please let us know if there are any additional questions or further clarifications we can provide to help improve the paper and gain your support.
> >
> > Thank you for your time and feedback!

---

> > > ### Comment · Reviewer_5UH3 · 2024-12-02
> > >
> > > Thank you for your detailed response. I appreciate the clarifications and additional analysis provided regarding sensitivity and computational complexity. I am retaining my score of 6.

---

> > > > ### Author Response · Authors · 2024-12-02
> > > > **Some clarity would be helpful**
> > > >
> > > > Since you raise "sensitivity to hyperparameters" as a weakness and also asked us a question about it, it would be useful to hear your response on whether you find our argument that (paraphrasing) "SURE is not sensitive to hyperparameters, compared to baselines" convincing or not. While we appreciate that you appreciate our rebuttal, your response somehow avoided saying anything. Since we ultimately have to revise the paper, hearing what you actually THINK about our rebuttal around sensitivity would be useful.

---

### Official Review · Reviewer_e2Uw · 2024-11-03

**Soundness:** 3
**Presentation:** 3
**Contribution:** 3
**Rating:** 6
**Confidence:** 3

**Summary:**

This paper proposes an approximate unlearning method that focuses on unlearning in the representation space rather than in the output space. Experiments demonstrate that the method has superior or comparable results compared to existing techniques.

**Strengths:**

1. This paper introduces a novel machine unlearning method for classification tasks (CIFAR-10, CINIC-10) via domain adversarial training, which shifts the focus from the output space to the representation space.

2. The performance gain from the proposed method is convincing.

3. The presentation is clear and easy to follow.

**Weaknesses:**

1. The paper compared its performance with 'Towards Unbounded Machine Unlearning, NIPS 2023'; ' by the MIA metric (User Privacy Settings). However, the proposed method should also be evaluated in removing biases (RB), Resolving Confusion (RC) settings. RB aims to achieve the highest possible error on the forgetting set, without hurting the error of the retain and the test set. It evaluates the performance of the proposed algorithm in unlearning the knowledge of the target domain while not hurting other's performance.  RC aims to resolve confusion between two classes that the original model suffers from due to a part of its training set being mislabelled. (settings from ''Towards Unbounded Machine Unlearning, NIPS 2023''）


2. Although the authors use CINIC-10 (a subset of ImageNet) to validate the effect of forgetting, the authors should still provide the effect of forgetting on the ImageNet-1K. There are many pre-trained models on ImageNet-1K and this study focuses on classification tasks, it is more convincing to test on ImageNet-1K to prove the real-world applicability of the proposed algorithm.


3. The title is ambiguous and should emphasize domain adversarial training rather than adversarial training to make it more clear, as domain adversarial training aims to improve the domain adaptation ability of the target model via an adversarial fashion, while adversarial training aims to improve the target model's robustness against adversarial attacks.

**Questions:**

1. The paper is mainly focused on classification tasks. I'm wondering if it can be extended to multi-modal tasks (image-text) and generation tasks (LLM, Diffusion)?

2.  Why does the reversal layer R(x)=x can helps to forget the representation & knowledge of D_f? Could the author provide more details about the optimization goal (2) on Page 6? (I see the optimization details in Appendix C, but still confused.  What is the relationship between L_d(G_d(R(G_f(x;\theta_f));\theta_d), 1(x\in D_f) ) and equation (7) (8) (9) in Page 14 & Appendix C?

---

> ### Author Response · Authors · 2024-11-28
> **rebuttal_reviewer_e2Uw**
>
> Thank you for your review. We like your suggestion to fix the title. This leaves only two weaknesses and three (related) questions. We think our responses below are, in our humble opinion, home runs and you may be reaching for that 10 score to ensure this paper gets published and shared with the world. You're likely wishing there were an 11. Honestly, though, we're happy that the question of the paper's acceptability for publication is settled. Right?! :)
>
> # WEAKNESSES: Addressed
>
> **Weakness: Should evaluate RB and RC metrics.** Thanks for this suggestion to look at the RB and RC benchmarks. The RB benchmark does not naturally fit into the usual privacy-theoretic framing of unlearning (where one aims to match retraining from scratch). On the other hand, RC can be interpreted as fitting into the standard framework, and so we tested SURE on RC and report the result below (tldr; we're tied with SOTA). Given our limited time, and the mismatch with RB, we focused solely on RC. It would be an interesting future direction to understand how to adapt SURE to the RB task.
>
> We evaluated the performance of our proposed unlearning method on the Resolving Confusion (RC) task using a ResNet-18 model trained on the CIFAR-10 dataset. We compared our method against all eight competing unlearning methods introduced in the paper. In our implementation, the forget set consists of a randomly selected subset (1%) of training samples from classes 0 and 1, which are intentionally mislabeled as the opposite class. Given that all samples in the forget set are mislabeled, the objective of unlearning in this context is to resolve the induced confusion. An unlearning method is successful if, after unlearning, the forget samples are correctly relabeled and the test and retain set performance is preserved.
>
> Results are now reported in Table 2 in Appendix F of the paper. Our findings demonstrate that SURE is the top-performing approach, alongside L1Sparse, yielding similar results for many metrics.
>
> **Weakness: Large benchmarks would be desirable.** Bigger is always better. But how does our evaluation compare to published work in this area? The largest dataset we have found studied in related unlearning papers is Tiny-ImageNet, potentially reflecting the challenges associated with scaling to larger datasets. Similarly, the architectures in related work have been limited to relatively smaller models like ResNet18 and, occasionally, ResNet50, with some studies using VGG-16 or ViT.
>
> Considering these findings and the rebuttal time constraints, we opted to tackle two additional configurations:
> 1. ResNet18 on Tiny ImageNet and  2.ViT-Small on CIFAR-10
>
> The results for these setups are presented in Appendix D, alongside the results for the CINIC-10 dataset. In short, SURE is the best or nearly so in everything. L1SPARSE is the only serious competition. What have we learned? In fresh, larger-scale benchmarks, our approach shows yet more promise. We would like to argue that unlearning needs NEW ideas and that our approach is both distinct and outperforms all prior methods in many scenarios (including the SOTA L1Sparse). Therefore, we think SURE will have an immediate impact on the subfield studying unlearning. Publishing this work will encourage other researchers to help explore the myriad ways this approach can be improved.
>
> **Weakness: Title should be "Using Domain Adversarial Training":** Fixed.
>
>
>
> # QUESTIONS: Answered
>
> **Can the method be extended beyond classifications to multi-modal and generation tasks?** We think that both these extensions are possible in principle but there is potentially a great deal of details to sort out to get things to work well. We focus on one modality and do extensive in-depth exploration on different metrics (including new ones we invented). Multimodality sounds like a great piece of future work for people who read our paper in the published proceedings of ICLR.
>
> **How does the Gradient Reversal Layer accomplish the stated objective?** We agree that the presentation of our training objective could have been clearer. In the main paper, we have tweaked the description, but continue to present the training objective in a way that is mimics how it is implemented in standard DL frameworks. However, we also provide a clear description of the underlying mathematical optimization problem in Appendix C, for those who are more comfortable with this approach. After defining the saddle point problem, we explain how the Gradient Reversal Layer simplifies the implementation. The GRL is a clever implementation trick to simplify training when part of the network is trained to minimize an objective and another part is trained to maximize the same objective. We won't recapitulate our explanation here, but we believe Appendix C should clarify your concerns.
>
> # Summary
> In summary, we believe we have offered ample additional evidence that our method is promising and is ready to be shared with the world with your approval.

---

> > ### Author Response · Authors · 2024-12-02
> > **followup-review_e2Uw**
> >
> > Dear Reviewer,
> >
> > We posted our responses to your comments a few days ago and wanted to kindly follow up. Have our replies addressed your concerns? Please let us know if there are any additional questions or further clarifications we can provide to help improve the paper and gain your support.
> >
> > Thank you for your time and feedback!

---

### Official Review · Reviewer_ezzj · 2024-11-04

**Soundness:** 3
**Presentation:** 3
**Contribution:** 2
**Rating:** 5
**Confidence:** 3

**Summary:**

The paper proposes a novel algorithm – SURE, for targeted unlearning. Inspired by work in domain adaptation, SURE treats a “forget set” and a validation set from the same distribution as data from two different domains, and optimizes the feature extractor to be s.t., the representations of the forget data are obfuscated from a discriminating domain classifier. Since SURE works directly on representation space, it prevents the stealing of information that may have remained in the latent space. The authors conduct experiments across a wide variety of utility metrics and compare their performance with many existing works.

**Strengths:**

- The paper is easy to understand.
- SURE demonstrates impressive results over existing works on a wide variety of metrics.
- SURE provides a stable training regimen for unlearning methods that are usually sensitive to hyper-parameters.

**Weaknesses:**

- The experiments are limited to CIFAR-10 and ResNet-18, albeit with a few results on CINIC-10. I believe that it’s important to observe the results on a wider variety of benchmarks, and more importantly models (like ViTs, etc) to ensure generality.
- Access to the validation set from the same distribution as the forget data: I am unsure how practically valid this assumption might be as the kinds of data people may want to remove are also primarily scarcely available, and if you trained and now want to unlearn on all of it (which seems plausible for current foundation models), it would not sit well with your current setup
- Novelty: The paper acknowledges that it heavily builds up on the DAN framework and applies it to the machine unlearning setup.

**Questions:**

- I would like to see results on different architectures and possibly more benchmarks.
- I would like to hear the authors' opinions on the practical implications of assuming access to the validation set.
- I would like to hear the authors' opinions on why SURE performs particularly (relatively) worse on RRT compared to other metrics.

---

> ### Author Response · Authors · 2024-11-28
> **rebuttal_reviewer_EZZJ**
>
> Thank you for your review. In summary, your review suggests our paper does not meet the bar for publication because our benchmarks are inadequate, you have concerns about the practicality of having a "validation set", and our approach may lack novelty by using existing domain adaptation technology (DANN). Our goal is to counter each of these arguments with such flair that, doing so, we convert you into our biggest advocate, eventually arguing with all the other reviewers that the paper is now a 10. Or some approximation of this reaction.
>
> # WEAKNESSES: Addressed
>
> **Weakness: Lack of variety in datasets and architectures.** We think we have made exciting innovations in our evaluations, especially around understanding the representation unlearning. But if we focus on the size and variety of datasets/architectures, how does our evaluation compare to published work in this area?
>
>
>
> The largest dataset we have found studied in related unlearning papers is Tiny-ImageNet, potentially reflecting the challenges associated with scaling to larger datasets. Similarly, the architectures in related work have been limited to relatively smaller models like ResNet18 and, occasionally, ResNet50, with some studies using VGG-16 or ViT.
>
> Considering these findings and the rebuttal time constraints, we opted to tackle two additional configurations:
> 1.ResNet18 on Tiny ImageNet and 2.ViT-Small on CIFAR-10
>
> The results for these setups are presented in Appendix D, alongside the results for the CINIC-10 dataset. In short, SURE continues to be SOTA in all the benchmarks on TinyImageNet (with the exception of RRT) and most benchmarks for ViT-small, except when L1Sparse performs a bit better. (We provide a detailed description of the results in the next comment, because we have passed the 5000 character limit.)
>
> We compared performance across the four top-performing methods: SURE (our method), L1Sparse, FineTune, and SCRUB. As before, the hyperparameters for each method were tuned using Bayesian optimization to find the best unlearning-utility tradeoff. These results cover several benchmarks introduced in the paper, addressing both random and partial class unlearning scenarios.
>
> What have we learned? In fresh, larger-scale benchmarks, our approach shows yet more promise. We would like to argue that unlearning needs NEW ideas and that our approach is quite distinct and outperforms all prior methods in many scenarios. We think SURE will have an immediate impact on the subfield studying unlearning. Publishing this work will encourage other researchers to help explore the myriad ways this approach can be improved.
>
> **Weakness. Assumption of an available validation set:** We apologize but we have likely caused some confusion here. In the random unlearning setting, the validation set is an "ordinary" validation set (i.e., equal in distribution to the training data), and so there's no practical concern here. In the partial class unlearning case, the validation set was restricted to the same class being forgotten. In practice, we also don't expect it to be a problem to have validation data for each class. In summary, there should not be a problem of scarcity for validation data. We think the composition of the validation set is an interesting place to focus in future work.
>
> **Weakness (Novelty). The paper heavily builds on DANN.** We don't believe that a novelty (treating unlearning as a domain adaptation problem, thus applying domain adversarial training) should end up being also a lack of novelty. To evaluate our novel approach, there was no need to invent an altogether-new approach. We would not be surprised if other groups that pick up this line of investigation look into new architectures, but evaluating the key idea did not require the additional complication.
>
> # QUESTIONS: Answered
>
> **On benchmarks: Answered above.**
>
> **On validation: Answered above.**
>
> **On slower runtime:** We acknowledge that SURE currently exhibits a slightly higher runtime compared to other unlearning methods, as reflected in both the absolute runtime (RT) and relative runtime (RRT) metrics (see the table posted in response to Reviewer 5UH3). Optimizing the runtime has, however, not been a priority of this project. We would argue that our results demonstrate that SURE shows enough promise to publish our findings, and hopefully inspire those with experience optimizing GPU workloads to join the effort.
>
> There are some promising signs about SURE's efficiency. In particular, SURE seems to have good "stability" to forget set size (in the sense of Section 4 of Fan et al. (2024)). Its sensitivity to hyperparameters seems to be good (Appendix E). In our experiment, standard BayesOpt delivered reliable results with SURE, in contrast to FineTuning and SCRUB which were very sensitive during HPT.
>
> # Summary
> SURE is a new approach to unlearning that outperforms all prior methods in many scenarios. We believe our work introducing this new approach is ready for publication.

---

> > ### Author Response · Authors · 2024-11-28
> > **further details about our additional experiments_revewer_EZZJ**
> >
> > For **Tiny-ImageNet**, our method outperforms all others across all benchmarks except for RRT (i.e., the time it takes to unlearn), which aligns with our findings for both CIFAR-10 and CINIC-10. In more detail, for random unlearning, BayesOpt failed to identify hyperparameters for FineTuning and SCRUB that achieved successful unlearning, meaning maintaining the desired unlearning-utility tradeoff. For class unlearning, FineTuning and SCRUB are more competitive, but the main competition is between our method (SURE) and L1Sparse, with our method outperforming in almost all benchmarks.
> >
> >
> > For **ViT-Small**, our method demonstrates solid performance across both scenarios. In the case of random unlearning, our method shows a small edge over L1Sparse. BayesOpt again failed to find hyperparameters for SCRUB that maintained the desired unlearning-utility tradeoff. For class unlearning, the results are more balanced: our method outperforms L1Sparse on 4 out of 7 benchmarks, while L1Sparse outperforms on the remaining 3. These findings highlight the robustness of our approach, while also demonstrating that pruning-based methods like L1Sparse can remain competitive in specific settings.

---

> > > ### Author Response · Authors · 2024-12-02
> > > **followup-reviewer_ezzj**
> > >
> > > Dear Reviewer,
> > >
> > >
> > > We posted our responses to your comments a few days ago and wanted to kindly follow up. Have our replies addressed your concerns? Please let us know if there are any additional questions or further clarifications we can provide to help improve the paper and gain your support.
> > >
> > >
> > > Thank you for your time and feedback!

---

### Meta-Review · Area_Chair_yrYA · 2024-12-18

**Metareview:**

The paper applied the idea from unsupervised domain adaptation to the targeted unlearning problem. While the method itself is directly adopted from the existing literature, DANN, its application to unlearning problem seems new. Various aspects of the unlearning have been considered and evaluated via extensive experiments and comparison between the existing baselines.

Some concerns were also raised during the reviewing period. (a) the availability of the "validation set", which needs to have the same distribution as the forget set, is one of the key component in the method, and (b) increased computation complexity. Both points need to be more clearly described and highlighted in the final version.

Despite the weakness, AC believes the application of domain adaptation idea to the unlearning problem can be regarded as an enough novelty to be accepted to ICLR. So, the final decision is accept.

**Additional Comments On Reviewer Discussion:**

The reviewers did not actively engage in the rebuttal process, so AC went through all their reviews/author rebuttal and the original submission. Followings are the key points from the reviews.

Reviewer ezzj raised the point about the validation set. When random unlearning is the target, assuming the availability seems to be fine, but for (partial) class unlearning, the subtlety should be more clearly addressed in the final version. Moreover, I believe most of the other baselines do not utilize such validation set, and the authors should argue this point clearly in the final version. (why making such assumption is not too critical.)

Reviewer e2Uw raised the point about the validity of the gradient reversal layer for unlearning. The authors provided their answer in Appendix C, but Ac believes this should also be argued in the main manuscript as well.

Reviewer 5UH3 asked about the computation complexity and the authors provided concrete comparison among the baselines. Please include this result in the final version as well.

---

### Decision · Program_Chairs · 2025-01-22

Accept (Poster)